# Combiner: Full Attention Transformer with Sparse Computation Cost

*Hongyu Ren[†], *Hanjun Dai[◇], *Zihang Dai[◇]
Mengjiao Yang[◇], Jure Leskovec[†], Dale Schuurmans[◇,‡], Bo Dai[◇]
[†]Stanford University, {hyren,jure}@cs.stanford.edu
[◇]Google Research, Brain Team, {hadai,zihangd,sherryy,schuurmans,bodai}@google.com
[‡]University of Alberta

## Abstract

Transformers provide a class of expressive architectures that are extremely effective for sequence modeling. However, the key limitation of transformers is their quadratic memory and time complexity $\mathcal{O}(L^2)$ with respect to the sequence length in attention layers, which restricts application in extremely long sequences. Most existing approaches leverage sparsity or low-rank assumptions in the attention matrix to reduce cost, but sacrifice expressiveness. Instead, we propose *Combiner*, which provides full attention capability in each attention head while maintaining low computation and memory complexity. The key idea is to treat the self-attention mechanism as a conditional expectation over embeddings at each location, and approximate the conditional distribution with a structured factorization. Each location can attend to all other locations, either via direct attention, or through indirect attention to *abstractions*, which are again conditional expectations of embeddings from corresponding local regions. We show that most sparse attention patterns used in existing sparse transformers are able to inspire the design of such factorization for full attention, resulting in the same sub-quadratic cost ($\mathcal{O}(L \log(L))$ or $\mathcal{O}(L\sqrt{L})$). Combiner is a drop-in replacement for attention layers in existing transformers and can be easily implemented in common frameworks. An experimental evaluation on both autoregressive and bidirectional sequence tasks demonstrates the effectiveness of this approach, yielding state-of-the-art results on several image and text modeling tasks.

## 1   Introduction

The Transformer [1] is a powerful neural network architecture that has demonstrated state-of-the-art performance in machine translation [2] and many other natural language processing (NLP) tasks via pretraining, using either unidirectional language modeling [3] or bidirectional language modeling [4–8]. It has also achieved excellent results in other domains like image recognition [9], code understanding [10], speech recognition [11], protein [12], music [13] and image [14] generative modeling. The core component of Transformer is the attention mechanism, which computes dependencies between all pairs of positions in a sequence. However, for a sequence of length $L$, the expressiveness of pairwise attention comes at a quadratic cost $\mathcal{O}(L^2)$ in both time and memory consumption. This makes the vanilla Transformer [1] prohibitive for applications that involve long sequences, including high-resolution images, protein sequences, or raw speech signals [15], where the sequence length $L$ is often larger than $10,000$ [14].

Recently, there have been several attempts to scale up attention to long sequences. A popular class of methods sparsifies the attention matrix with different sparsity patterns, including local

---

*indicates equal contribution. The work was completed during HR's internship at Google Brain.

35th Conference on Neural Information Processing Systems (NeurIPS 2021).

window [16, 17], local+stride [14], log-sparse [18], axial [19, 20], or learnable patterns through hashing [21] or clustering [22]. Sparse attention enjoys sub-quadratic cost, but is lossy in capturing all-pair relationships. Generally, sparse attention requires more layers [14, 20, 23] to achieve full autoregressive or bidirectional dependencies (or receptive fields [20]) for each location in a long sequence.

Alternatively, another line of research has tried to achieve scalability with an explicit low-rank assumption [24, 25] on the attention matrix or by using explicit feature maps of some kernels [26]. However these explicit low dimensional approximations might be too restricted for the potentially full rank attention matrix, which uses exponential kernels that are effectively infinite dimensional [27]. The Performer [28] is among the first works that attempts to approximate regular full-rank attention with the random feature trick [29]. However such random-feature based approaches [30] require many more bases to better approximate the exponential kernel [27], and empirically we found it produces inferior results in some sequence modeling tasks, such as density estimation.

In this paper we propose *Combiner*, a drop-in replacement for the vanilla quadratic attention mechanism with sub-quadratic computation and memory cost. Combiner still achieves full attention capability within each head of Multi-Head Attention, unlike approaches that adopt sparse or low-rank approximations. As we will discuss, the standard attention computed at each location can be seen as the conditional expectation of the value embeddings at all feasible locations given the current location. Based on such an understanding, Combiner explicitly approximates the conditional distribution in through a structured factorization of the probability space. Specifically, given a location $x$, the probability of attending to location $y$ can be either *directly* calculated via the query vector of $x$ and key vector of $y$, or *indirectly* through a local *abstraction* where $x$ first attends to the key vector that represents a group of locations containing $y$, and multiplying the probability of choosing $y$ within that group. We refer to this model as Combiner since the conditional distributions in attention become a combination between several local attentions and direct attentions. This structured decomposition enables Combiner to take existing sparse attention patterns and convert them into corresponding design choices for probability factorizations that achieve full attention. As shown in Figure 1, Combiner achieves full attention with the same asymptotic complexity as sparse variants. Combiner can be easily implemented in most existing deep learning frameworks without the need for specialized hardware implementation, and is GPU/TPU friendly. In fact, both the fixed and learnable sparse attention patterns from many existing Transformer variants [14, 18, 20, 22] can be enhanced with such structured factorizations, with *the same order* of time or memory cost.

We validate Combiner on both autoregressive and bidirectional sequence modeling tasks over a variety of domains including text and images. We show that Combiner can achieve better perplexity and accuracy when using the same transformer architectures while being much faster in terms of runtime, and achieves state of the art performance on density estimation on standard datasets CIFAR-10 (2.77 bits/dim) and ImageNet-64 (3.42 bits/dim), as well as the Long-Range Arena [31]. The implementation of Combiner can be found at https://github.com/google-research/google-research/tree/master/combiner.

## 2 Attention as Conditional Expectation

In this section, we revisit the formulation of the standard Transformer [1] from the perspective of conditional expectation, which inspires the derivation of Combiner.

Without loss of generality, we use a single sequence in the self-attention scenario. Given a sequence of $L$ embeddings $X = [x_1, x_2, \ldots, x_L]$, where $X \in \mathbb{R}^{L \times d}$ and each embedding $x_i \in \mathbb{R}^d$ is a $d$-dimensional vector, the core component of Transformer is the multi-head attention, where each head $h$ is a scaled dot-product attention:

$$A_h(X) = \texttt{softmax}\left(\frac{Q_h}{\sqrt{d}} K_h^\top\right) V_h, \left\{Q_h = X W_h^Q, K_h = X W_h^K, V_h = X W_h^V\right\} \in \mathbb{R}^{L \times d}, \quad (1)$$

and the attention vector from each head $A_h(X)$ is concatenated and projected:

$$\texttt{MultiHeadAttn}(X) = [A_1(X), A_2(X), \ldots, A_H(X)] W^o, W^o \in \mathbb{R}^{Hd \times d}. \quad (2)$$

Here $H$ is the total number of heads per Transformer layer. In this paper, we focus on how to approximate full attention within *each* head of multi-head attention. For ease of notation, we drop the head index $h$ whenever possible, and use lower-case letters $x_i, q_i, k_i, v_i \in \mathbb{R}^d$ to denote rows in

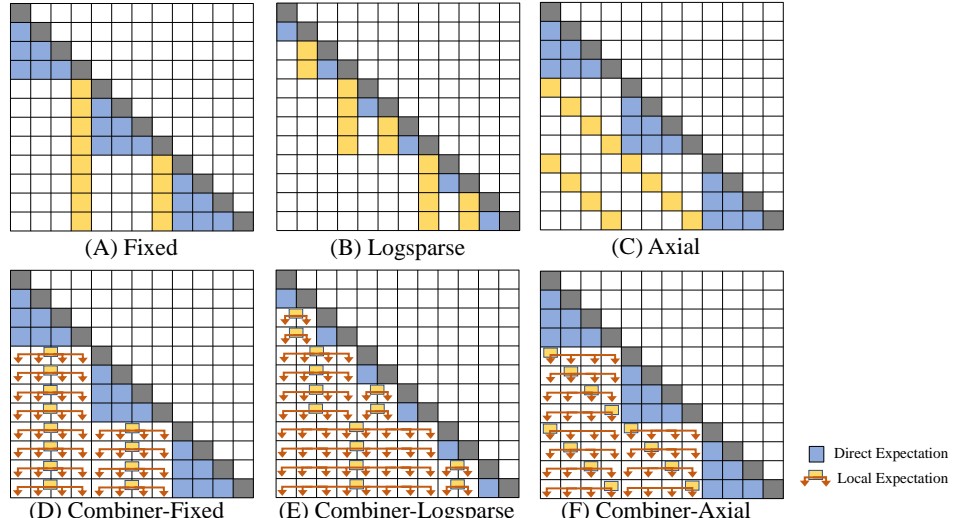

| (A) Fixed | (B) Logsparse | (C) Axial |
|---|---|---|
| (D) Combiner-Fixed | (E) Combiner-Logsparse | (F) Combiner-Axial |

Direct Expectation
Local Expectation

Figure 1: Attention matrices of several instantiations of Combiner in the autoregressive setting. We transform several sparse attention patterns: Fixed (A) [14], Logsparse (B) [18] and Axial (C) [20] to Combiner-Fixed (D), Combiner-Logsparse (E) and Combiner-Axial (F). Combiner approximates the conditional expectation (3) with a combination of direct expectation (blue) and local expectation (yellow). Our instantiations (D)(E)(F) achieves full attention with the same sub-quadratic complexity.

$X, Q, K, V$ respectively, which corresponds to a location $i$ in the original sequence of length $L$. We use $[n]$ to denote the set of positive integers $\{1, 2, \ldots, n\}$.

For a position $i \in [L]$, the attention formulation (1) can be viewed as conditional expectation of rows in $V$. Specifically, since softmax outputs a probability distribution, we can rewrite (1) as

$$A(x_i) = \mathbb{E}_{p(j|i)} [v_j], \qquad p(j|i) = \frac{1}{Z(x_i)} \exp\left(\frac{q_i}{\sqrt{d}} k_j^\top\right), \qquad (3)$$

where $p(j|i)$ denotes the conditional probability at position $j$ given the token at position $i$ and the partition function $Z(x_i) = \sum_{j \in \Omega_i} \exp\left(\frac{q_i}{\sqrt{d}} k_j^\top\right)$ over support $\Omega_i$. The support $\Omega_i$ of $p(j|i)$ defines the set of valid locations that the $i$-th token can attend to. For instance, the support set in autoregressive language modeling (LM) consists of all previous tokens, i.e., $\Omega_i^{\text{LM}} = [i]^2$; in masked language modeling (MLM) the support consists of all tokens in the sequence, i.e., $\Omega_i^{\text{MLM}} = [L]$. That is, $\Omega_i^{\text{LM}}$ and $\Omega_i^{\text{MLM}}$ represent the full attention capability respectively in the LM and MLM setting.

## 3   Combiner: Full Attention via Structured Conditional Expectation

The complexity of $p(j|i)$ is the bottleneck of the computation for $A(x_i)$. Generally, in existing sparse transformers, the support of $p(j|i)$ is sparsified to reduce the computation and memory complexity, e.g., $\Omega_i^{\text{Sparse}} \subsetneq \Omega_i^{\text{LM}}$ for LM and $\Omega_i^{\text{Sparse}} \subsetneq \Omega_i^{\text{MLM}}$ for MLM, but this can lead to either reduced capacity or limited applicability. We defer detailed discussion of the full capacity of the model to Appendix A. In this section we introduce the Combiner, which achieves $\Omega_i^{\text{Combiner}} = \Omega_i^{\text{LM}}$ for LM and $\Omega_i^{\text{Combiner}} = \Omega_i^{\text{MLM}}$ for MLM, while still maintaining sub-quadratic computation and memory cost. Below we denote $\Omega_i$ as the support for full attention if there is no ambiguity or need to distinguish between LM or MLM. We introduce the main design framework in Section 3.1 and possible parameterizations in Section 3.2. Then in Section 3.3 we analyze the trade-off of Combiner.

### 3.1   Local Factorization for Conditional Expectation

The main idea of Combiner is to exploit a hierarchical structure for conditional probability modeling in (3), which provides the opportunity for reducing computation complexity while maintaining the

---

$^2$Following the conventional implementation, the input sequence will be "right-shifted" so that the position $i$ can attent to itself in LM setting.

same support. Specifically, we introduce support variables $\Omega_i^r$, for $r = 0, \ldots, n_i$ and $i \in [L]$. The support variables are disjoint, *i.e.*, $\Omega_i^r \cap \Omega_i^s = \emptyset, \forall r \neq s$, and $\cup_{r=0}^{n_i} \Omega_i^r = \Omega_i$. Then we can factorize $p(j|i)$ as

$$p(j|i) = \sum_{r=0}^{n_i} p(j, \Omega_i^r|i) = \sum_{r=0}^{n_i} p(j|\Omega_i^r, i)p(\Omega_i^r|i) = \textcolor{red}{p(j|\Omega_i^{r_j}, i)}p(\Omega_i^{r_j}|i), \tag{4}$$

where $r_j$ denotes the index of the support to which $j$ belongs. The last equation arises from the fact that the $\Omega_i^r$ are disjoint from each other ($\Omega_i^r \cap \Omega_i^s = \emptyset, \forall r \neq s$). Therefore, there is only one support, $\Omega_i^{r_j}$, containing $j$. The remaining terms, where $j \notin \Omega_i^r$ for $r \neq r_j$, are all zero since $p(j|\Omega_i^r, i) = 0$.

Furthermore, assume $\Omega_i^{r_j}$ is a sufficient statistic, *i.e.*, $j$ and $i$ are independent given $\Omega_i^{r_j}$, we obtain

$$p(j|i) = \textcolor{red}{p(j|\Omega_i^{r_j})}p(\Omega_i^{r_j}|i). \tag{5}$$

Given the partition $\{\Omega_i^r\}_{r=0}^{n_i}$, the attention form in (3) can be rewritten as

$$A(x_i) = \mathbb{E}_{p(j|i)}[v_j] = \sum_{r=0}^{n_i} \sum_{j \in \Omega_i^r} p(j, \Omega_i^r|i) v_j \tag{6}$$

$$= \underbrace{\sum_{j \in \Omega_i^0} \tilde{p}(j|i)v_j}_{\text{direct expectation}} + \sum_{r=1}^{n_i} p(\Omega_i^r|i)\underbrace{\left( \sum_{j \in \Omega_i^r} p(j|\Omega_i^r)v_j \right)}_{\text{local expectation}}, \tag{7}$$

where we consider direct attention in partition $\Omega_i^0$ and apply the local factorization (5) to the partition $r = 1, \ldots, n_i$. Here $\tilde{p}(j|i) \propto p(j|i)$ but with different normalization constants, which will be explained below. We refer to this model as *Combiner* since the structured attention (7) combines the direct expectation of $\Omega_i^0$ and multiple local expectations via $p(j|\Omega_i^r)$ and $p(\Omega_i^r|i)$ to form the final conditional expectation.

Equivalently, we can also rewrite the structured attention (7) as

$$A(x_i) = \sum_{j \in \Omega_i} \underbrace{\left[ \mathbb{I}(j \in \Omega_i^0)\tilde{p}(j|i) + \sum_{r=1}^{n_i} \mathbb{I}(j \in \Omega_i^r)p(j|\Omega_i^r)p(\Omega_i^r|i) \right]}_{\text{the new effective conditional probability } q(j|i)} v_j, \tag{8}$$

where $\mathbb{I}(\cdot)$ is a binary indicator function. After reordering, one can see from (8) that we obtain the effective conditional probability $q(j|i)$ that tries to approximate the original $p(j|i)$. Each probability term depends on both current location $i$ and other location $j$, and the expectation is still obtained with respect to a valid conditional probability (non-negative and sums up to 1 over $\Omega_i$).

**Requirement for Sub-quadratic Cost.** We can immediately see the benefit of this formulation from the fact that the *local expectation* in (7) is independent of the position $i$. The full dependence is achieved via the multiplier $p(\Omega_i^r|i)$ where $j \in \Omega_i^r$. If we can design the local factorization such that:

1. the order of number of terms in (7) for $p(\cdot|i)$, $\forall i \in [L]$: $\sum_{i=1}^{L}(n_i + |\Omega_i^0|)$ is sub-quadratic; and
2. let $\mathcal{U} = \{\Omega_i^r\}_{i \in [L], r \in [1, n_i]}$ be the unique set of partitions used for local expectation calculation, then the order of $|\mathcal{U}|$ (*i.e.*, the number of unique partitions in $\mathcal{U}$) is sub-quadratic;
3. the order of total number of unique calculations of local expectation across all locations in (7), $\sum_{\Omega \in \mathcal{U}} |\Omega|$ is sub-quadratic;

then one can see that the overall computation and memory cost will be sub-quadratic with full attention support $\Omega_i^{\text{Combiner}} = \Omega_i$, $\forall i \in [L]$. We will discuss in detail in Section 4 how to instantiate such a principle by drawing inspiration from existing sparse transformers, and how to convert them into a full attention model almost for free with identical asymptotic complexity.

**Remark (Further Hierarchical Decomposition):** We introduce the local decomposition with a one layer partition of support of $p(\cdot|i)$ for simplicity. In fact, such local decompositions can be stacked further, which introduces a partition tree. Specifically, we can further partition $\Omega_i^r$ with disjoint subsets $\left\{\Omega_i^{rk}\right\}_{k=1}^{n_r}$, and consider local decomposition $p(j, \Omega_i^r|i) = p(j|\Omega_i^{rk_j}, i)p(\Omega_i^{rk_j}|\Omega_i^r, i)p(\Omega_i^r|i)$, where $k_j$ is the index of sub-region which $j$ belongs to. Thus, we obtain a hierarchical decomposition of $p(j|i)$, which can also be plugged to (6) and yield a new full attention formulation.

## 3.2 Parameterizing Conditional Probabilities

While we obtained a possible way to speed up the standard Transformer via a combination of direct expectation and local expectations, it is also important to have an efficient design choice for the probability terms in (7), namely $\tilde{p}(j|i)$ from direct expectation, $p(j|\Omega_i^r)$ from local expectation and $p(\Omega_i^r|i)$ for $r \in [1, n_i]$. For simplicity we use the scaled dot-product, which means that we will associate positions $i, j$ and variable sets $\Omega_i^r$ with the corresponding embedding representation, and thus the probability is proportional to the exponential of the embedding inner products. Specifically:

- $\tilde{p}(j|i)$: As this term is for the direct expectation, we can let $\tilde{p}(j|i) \propto \exp(\frac{q_i}{\sqrt{d}}k_j^\top)$, which is the same as vanilla attention (3) but with different normalizations, which will be explained in Equation 9.
- $p(\Omega_i^r|i)$: This term aims to capture the joint event probability, *i.e.*, $p(\Omega_i^r|i) \propto \exp\left(\frac{q_i}{\sqrt{d}}k_{\Omega_i^r}^\top\right)$. Thus the design choice of $k_{\Omega_i^r}$ should make an *abstraction* of the corresponding support $\Omega_i^r$. We find $k_{\Omega_i^r} = \max \text{pooling}_{j \in \Omega_i^r} k_j$ already provides good empirical results without introducing additional parameters; we can also use DeepSets [32] to obtain such abstraction.
- $p(j|\Omega_i^r)$: This term is the probability of getting $j$ within this local span $\Omega_i^r$. We make $p(j|\Omega_i^r) \propto \exp\left(\frac{q_{\Omega_i^r}}{\sqrt{d}}k_j^\top\right)$, where we use max pooling or DeepSets over $\{q_j\}_{j \in \Omega_i^r}$ to obtain $q_{\Omega_i^r}$ similarly.

**Normalizing Probability Terms.** The terms in each local expectation $p(j|\Omega_i^r)$, $\forall j \in \Omega_i^r$ can be normalized within the local span; the direct expectation $\tilde{p}(j|i)$ and the terms in $p(\Omega_i^r|i)$ should be normalized together,

$$Z(x_i) = \sum_{j \in \Omega_i^{(0)}} \exp\left(\frac{q_i}{\sqrt{d}}k_j^\top\right) + \sum_{r=1}^{n_i} \exp\left(\frac{q_i}{\sqrt{d}}k_{\Omega_i^r}^\top\right), \tag{9}$$

and $Z(x_i)$ is the normalizing constant when calculating $\tilde{p}(j|i)$ and $p(\Omega_i^r|i)$.

## 3.3 Trade-offs in Combiner

Combiner achieves full attention with reduced cost without making explicit sparsity or low-rank assumptions over the attention matrix. However this efficiency gain is not free. In this section we discuss the limitations of the simplification made by Combiner, and provide a simple workaround.

**Structured Attention Approximation.** We obtain the local decomposition (5) under the *conditional independence* assumption. Therefore, the *local expectation* in (7) is independent of the position $i$, this suggests that any two locations $i_1$ and $i_2$ with $\Omega_{i_1}^r = \Omega_{i_2}^r = \Omega$ would have linearly dependent attention scores over the region $\Omega$. Formally, the probabilities formed by the effective conditional distribution $\vec{a}(\Omega)_{i_1} = \left[q(j_1|i_1), q(j_2|i_1), \ldots, q(j_{|\Omega_{i_1}^r|}|i_1)\right] = \frac{p(\Omega_{i_1}^r|i_1)}{p(\Omega_{i_2}^r|i_2)}\vec{a}(\Omega)_{i_2}$. In other words, the rank of the sub-matrix over the same partition in the resulting attention matrix is 1, therefore, the attention matrix is locally low-rank based on the partition. On the other hand, the *direct expectation* fully attends to each position in sub-support $\Omega_0$, which ensures the full-rank block. These two attention schemes make the attention matrix of Combiner structured. Compared with the low-rank approximation for attention [26, 28, 30], which is inspired from random features [29] in the kernel community, a structured approximation that exploits both the locally low-rank and full-rank blocks has been proved more powerful theoretically and empirically in large-scale kernel machines [27].

**Improving Expressiveness Using a Mixture Model.** One way to further improve the expressiveness of the local factorization is to use a mixture model. This idea is adapted from the mixture of softmaxs [33] to obtain high-rank softmax layer in language modeling. Let $\omega$ be a certain partition of the support (*i.e.*, collection of $\Omega_i^r$) of $\Omega_i$, then one can easily use $A(x_i) = \frac{1}{M}\sum_{m=1}^{M} A(x_i; \omega_m)$ to compute the attention, where each component of the mixture $A(x_i; \omega_m)$ is the term (7) using a specific factorization plan $\omega_m$. Empirically we find two components are already sufficient to improve performance.

## 4 Combiner Instantiations

In this section we show several local factorization schemes satisfying the requirements in Section 3.1. As we will see, Combiner is able to convert several sparse transformers [14, 18, 20–22] into full

attention, with the same order of computation and memory consumption. One can also design other factorization patterns, which can be easily instantiated in Combiner.

## 4.1 Combiner-Fixed

The Sparse Transformer [14] is one of the most representative variants that can achieve $\mathcal{O}(L\sqrt{L})$ computation and memory cost with sparse attention. Here we show how to convert this fixed pattern proposed in [14] (Figure 1(A)) into a factorization plan, and instantiate a full attention variant named the *Combiner-Fixed* (Figure 1(D)).

In the fixed-sparse attention, the support is $\Omega_i^{\text{sparse MLM}} = \{j : j \bmod s = 0\} \cup \{j : j \equiv i\,(\text{div } s)\}$ where $s$ is a hyper-parameter, div is integer division, and $j \equiv i\,(\text{div } s)$ denotes that the quotients of $i$ and $j$ w.r.t. $s$ are the same. In the autoregressive case, $\Omega_i^{\text{sparse LM}} = \Omega_i^{\text{sparse MLM}} \cap [i]$. Please refer to Figure 1(A) for an illustration of the LM version.

Our design of $\omega_{\text{fixed}}^{\text{MLM}}$ has the following form:
$$\Omega_i^0 = \{j : j \equiv i\,(\text{div } s)\}, \Omega_i^r = \{j : j \text{ div } s = r, j \notin \Omega_i^0\}, \forall r \in [L \text{ div } s], \forall i \in [L] \qquad (10)$$
where each *local expectation* is performed in each span of size $s$, and there are totally $L$ div $s$ spans across all locations. For each position $i \in [L]$, there are $(s + (L \text{ div } s))$ terms in (7) ; the local expectation has $(L \text{ div } s)$ terms . The overall complexity is $\mathcal{O}(L \cdot (s + 2(L \text{ div } s)))$. The optimal $s$ is $\mathcal{O}(\sqrt{L})$, and we can achieve $\mathcal{O}(L\sqrt{L})$ computation and memory complexity, which is the same as [14] but here we gain full attention capability in each attention head. For the LM case, we can simply have $\omega_{\text{fixed}}^{\text{LM}} : \{\Omega_i^r \cap [i] \mid \Omega_i^r \in \omega_{\text{fixed}}^{\text{MLM}}\}$, which has the same $\mathcal{O}(L\sqrt{L})$ optimal complexity.

## 4.2 Combiner-Logsparse

The Logsparse Transformer is proposed in [18] and can theoretically achieve $\mathcal{O}(L \log L)$ cost. The general idea is to make the size of support $\Omega_i^{\text{sparse}}$ no larger than $\lceil \log_2 i \rceil$. For the ease of notation, we first define $\text{bits}(n) = [b_1, b_2, \ldots, b_{\lceil \log_2 n \rceil}]$ to be the binary representation of integer $n$, with $b_t \in \{0, 1\}$ the coefficient of basis $2^t$. Thus we have $n = \sum_{t=1}^{\lceil \log_2 n \rceil} b_t * 2^t$. One of the possible design choices to make Logsparse in the LM case is $\Omega_i^{\text{sparse LM}} = \left\{ \text{suff}_t := \sum_{\tau=t}^{\lceil \log_2 i-1 \rceil} b_\tau * 2^\tau \right\}_{t=1}^{\lceil \log_2 i-1 \rceil} \cup \{i\}$, *i.e.*, attend to the location indices that equal to the suffix sum of the weighted $\text{bits}(i - 1)$, as well as location $i$ itself. This serves as our base sparse version as shown in Figure 1(B).

To exploit this scheme in the Combiner framework, we can define $\lceil \log_2 n \rceil$ non-overlapping supports, where $\Omega_i^r = [\text{suff}_r] \setminus [\text{suff}_{r+1}]$ with the boundary case $[\text{suff}_{\lceil \log_2 i-1 \rceil+1}] = \emptyset$. Note that for the ease of notation, some of the $\Omega_i^r$ are empty which will be ignored. In this case, the direct attention set $\Omega_i^0$ includes $\{i\}$, as well as $\{i - 1\}$ when $i$ is an even number. Such a factorization leads to *Combiner-Logsparse*, as shown in Figure 1(E). From the Figure, we observe that in total we will have span summaries for every $2, 4, 8, \ldots, 2^{\lfloor \log_2 L \rfloor}$ locations, resulting in total $\sum_{t=1}^{\lfloor \log_2 L \rfloor} \lfloor \frac{L}{2^t} \rfloor$ or $\mathcal{O}(L)$ summaries. Each location $i$ will select at most $\mathcal{O}(\log(i))$ non-overlapping spans to cover the full support $\Omega_i$, and thus, the total cost will be $\mathcal{O}(L \log L)$. We leave the design of MLM case to Appendix B.

## 4.3 Combiner-Axial

The Axial Transformer [20] builds the attention along each axis of the input data. Without loss of generality, we focus on 2D case where the input sequence is reshaped into a matrix of size $n \times m = L$. Specifically, the location $i$ in original sequence will be in $row_i = (i - 1) \text{ div } m + 1$ and $col_i = (i - 1) \bmod m + 1$. We show how to simply enable full attention with factorization on 2D matrix, hence *Combiner-Axial*.

The sparse axial has $\Omega_i^{\text{sparse MLM}} = \{j : j - 1 \equiv i - 1 (\bmod m)\} \cup \{j : j - 1 \equiv i - 1 (\text{div } m)\}$, and $\Omega_i^{\text{sparse LM}} = \Omega_i^{\text{sparse MLM}} \cap [i]$, which all have at most $O(m + n)$ entries for each $i$, as illustrated in Figure 1(C). We propose several factorization schemes to make it an attention with full support.

- $\omega_{\text{axial-vertical}}^{\text{LM}}$: $\Omega_i^0 = \Omega_i^{\text{sparse LM}}$, and $\Omega_i^r = \{j : j \equiv r (\bmod m)\} \cap [i - col_i]$, for $r \in [m] \setminus \{col_i\}$. As depicted in Figure 2(A), $\Omega_i^r$ corresponds to the column $r$ above $row_i$, where we use max pooling to

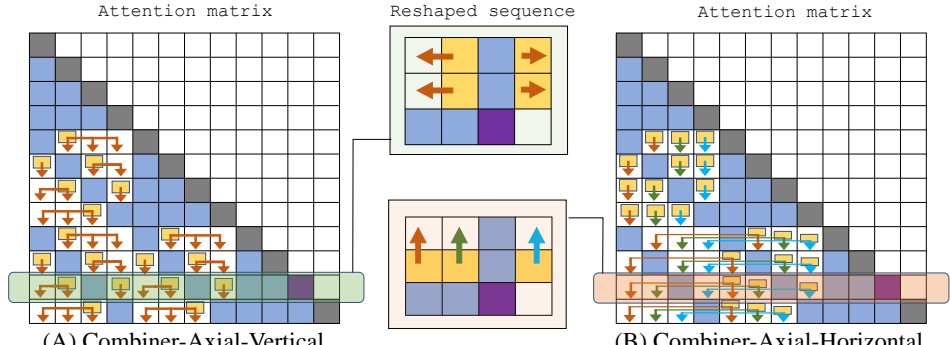

Figure 2: Attention matrices and sequence being attended (e.g., a 3x4 image) of vertical and horizontal variants of Combiner-Axial. Blue and yellow correspond to direct and local attention respectively for location $i$ (purple). Locations connected by arrows correspond to the same support $\Omega^r$.

obtain the abstraction. To obtain such abstraction for all the locations, we can leverage the `cummax` operator for each column to efficiently obtain the prefix-max.

- $\omega_{\text{axial-horizontal}}^{\text{LM}}$: similar as $\omega_{\text{axial-vertical}}$ except that each $\Omega_i^r$ summarizes the row $r$ before $row_i$ and excludes $col_i$ (Figure 2(B)).

- $\omega_{\text{axial-rowmajor}}^{\text{LM}}$: $\Omega_i^0 = \{j : j - 1 \equiv i - 1(\text{div } m)\} \cap [i]$, *i.e.*, elements in the same row are directly attended, while $\Omega_i^r = \{j : j \equiv r(\text{div } m)\} \cap [i - col_i]$ captures the rows before $row_i$. This structure is similar to Combiner-Fixed, except for the way that the *abstraction* (and thus the *local expectation*) is computed. Combiner-Fixed computes the *abstraction* only based on $r$ of partition $\Omega_i^r$, where $\omega_{\text{axial-rowmajor}}$ depends on both $r$ and the column $col_i$ (Figure 1(F)).

In all cases above, the cost is similar to the Axial Transformer [20], which is $O(L\sqrt{L})$ if we reshape the sequence to a 2D matrix with $n, m = O(\sqrt{L})$. We defer the MLM case to Appendix C.

## 4.4 Combiner-Learnable

Inspired by the Reformer [21] and Routing Transformer [22], we can also learn the factorization plan $\omega$ from the data. We illustrate this with Routing Transformer and provide a way to enable full attention in Routing Transformer following the Combiner principle.

For a specific layer, suppose we have a learned disjoint region (or cluster in Routing Transformer) $\{\Omega^r\}_{r=1}^n$ where $\cup_r \Omega^r = [L]$. In Routing Transformer, we simply have $\Omega_i^{\text{sparse MLM}} = \Omega^{r_i}$ where $\Omega^{r_i}$ denotes the region where position $i$ belongs to. To define the Combiner factorization, we let

$$\omega_{\text{routing MLM}} : \Omega_i^0 = \Omega^{r_i}, \quad \Omega_i^r = \Omega^r \setminus \Omega_i^0, \quad \forall r \in [n_i]. \tag{11}$$

Note that $n_i = n$ (*i.e.*, number of learned clusters) for all locations. The above factorization can only work for MLM. LM requires the following definition:

$$\omega_{\text{routing LM}} : \Omega_i^0 = \Omega^{r_i} \cap [i], \quad \Omega_i^r = (\Omega^r \setminus \Omega_i^0) \cap [i], \quad \forall r \in [n_i]. \tag{12}$$

In general, both LM and MLM can have sub-quadratic cost when $n = O(\sqrt{L})$. However, routing variants (including the Routing Transformer) require a `gather` operation, which can be slow on TPUs (see illustration in Appendix D).

## 5 Experimental Evaluation

We evaluate Combiner with different full attention patterns on both autoregressive and bidirectional sequence modeling tasks, covering a wide range of input data from images to texts. All tasks considered involve long sequences for up to 12,000 in length, some of which prevent the applicability of the vanilla transformer. We compare Combiner with state-of-the-art Transformers. We also perform a series of ablation studies where all of the models being compared use the *exact same architecture* that only differ in the attention module, avoiding individual tricks employed in the original works (*e.g.*, using both learnable and fixed patterns in Routing Transformer [22]). Details to reproducing all experimental results can be found in Appendix E.

Table 1: Ablation results in Bits per Dimension (Bits/Dim) on CIFAR-10 and ImageNet-64.

| Model | Layers | CIFAR-10 | ImageNet-64 |
|---|---|---|---|
| Reformer [21] | 6 | - | 3.740 |
| Performer [28] | 6 | 3.335 | 3.719 |
| Logsparse [18] | 6 | 4.253 | 4.351 |
| Combiner-Logsparse (Ours) | 6 | 3.366 | 3.795 |
| Fixed [14] | 6 | 3.408 | 3.696 |
| Combiner-Fixed (Ours) | 6 | 3.321 | 3.654 |
| Axial [20] | 6 | 3.666 | 4.032 |
| Combiner-Axial (Ours) | 6 | 3.050 | **3.585** |
| Combiner-Mixture (Ours) | 6 | **3.040** | **3.585** |
| Reformer [21] | 12 | - | 3.710 |
| Performer [28] | 12 | 3.310 | 3.636 |
| Routing Transformer [22] | 12 | 2.950 | - |
| Combiner-Mixture (Ours) | 12 | **2.885** | **3.504** |

## 5.1 Autoregressive Sequence Modeling

In this subsection, we first perform density estimation on text and image using Combiner.

### 5.1.1 Language Modeling

For language modeling, we focus on the Wiki-40B-En dataset [34], which consists of clean Wikipedia pages in English. We use a sentence piece model with vocabulary size 32K to tokenize the text and measure the perplexity at the sentence piece level. To ensure fair comparison, all models being compared again have the same number of layers and hidden sizes, are are implemented under the same code base.

Table 2 shows the results of the comparison. As we can see, under 2k sequence length, Combiner variants are consistently better than their corresponding baselines, and are very close to the standard Transformer. When sequence length goes to 8k, the standard Transformer runs out of memory, whereas Combiner continues to achieve improved perplexity, surpassing the result of Transformer-2k. If we further use DeepSets to calculate the summarization terms $q_{\Omega_i^r}$ and $k_{\Omega_i^r}$, we may further achieve lower perplexity as shown in Table 3.

Table 2: LM Perplexity on Wiki-40B (Main).

| Model | Perplexity |
|---|---|
| Transformer-2k [1] | 17.26 |
| Performer-2k [28] | 19.66 |
| Routing-2k [22] | 20.85 |
| Fixed-2k [14] | 18.04 |
| Combiner-Fixed-2k (Ours) | 17.70 |
| Axial-2k [20] | 20.82 |
| Combiner-Axial-2k (Ours) | 17.56 |
| Combiner-Fixed-8k (Ours) | 16.60 |
| Combiner-Axial-8k (Ours) | **16.49** |

Table 3: LM Perplexity on Wiki-40B (Ablation).

| Model | Perplexity |
|---|---|
| Transformer-2k [1] | 17.26 |
| Combiner-DeepSets-Max-8k (Ours) | **16.29** |
| Combiner-DeepSets-Mean-8k (Ours) | 16.48 |
| Combiner-Max-8k (Ours) | 16.60 |
| Combiner-Mean-8k (Ours) | 16.54 |

### 5.1.2 Image Generative Models

**CIFAR-10.** We first perform a sanity check where we compare sparse attention baselines against Combiner with full attention under the *same architecture* on the CIFAR-10 dataset. The sequence length is 3072. For all the methods, we use a same 6-layer transformer with 8 attention heads and 512 embedding dimensions. We train all models for 500k iterations using batch size 32 on TPU v2. As shown in Table 1, given the same model architecture, Combiner-X performs significantly better than the base model X under the bits per dimension (BPD) metric on the 10,000 test images. In particular, Combiner significantly decreases BPD by 0.887, 0.087, and 0.626 compared to the base models Logsparse, Fixed and Axial, respectively. Note that all of the Combiner variants achieve better performance than the best of the base models. This demonstrates the advantage of Combiner over the baselines given the same 6-layer architecture. We observe a similar trend under a 12-layer architecture.

Table 4: Bits per Dimension (Bits/Dim) on CIFAR-10 and ImageNet-64.

| CIFAR-10 | Bits/Dim |
|---|---|
| PixelCNN [15] | 3.03 |
| PixelCNN++ [36] | 2.92 |
| Image Transformer [16] | 2.90 |
| PixelSNAIL [37] | 2.85 |
| Sparse Transformer [14] | 2.80 |
| **Combiner-Axial (ours)** | **2.77** |

| ImageNet 64x64 | Bits/Dim |
|---|---|
| PixelCNN [15] | 3.57 |
| Parallel Multiscale [38] | 3.70 |
| Glow [39] | 3.81 |
| SPN [40] | 3.52 |
| Sparse Transformer [14] | 3.44 |
| Axial Transformer [20] | 3.44 |
| Routing Transformer [22] | 3.43 |
| **Combiner-Axial (ours)** | **3.42** |

Following the 128-layer architecture in Child et al. [14], we apply Combiner-Axial and achieve state-of-the-art performance, 2.77 BPD on CIFAR-10, as listed in Table 4. We run all of the models in Table 4 without data augmentation [35].

**ImageNet-64.** We also evaluate performance under the autoregressive setting on ImageNet-64, where sequence length is 12,288. We first perform the same analysis as CIFAR-10 and compare Combiner-X with the baselines using the same model architecture. As shown in Table 1, Combiner consistently outperforms the baselines with the same attention pattern. We further apply Combiner-Axial to a 30-layer Transformer, which achieves state-of-the-art performance on density estimation on ImageNet-64, demonstrating the effectiveness of full attention achieved by Combiner.

## 5.2 Bidirectional Sequence Modeling

Besides autoregressive tasks, we also evaluate Combiner on a set of standard bidirectional tasks to show the general applicability of the method.

### 5.2.1 Long-Range Arena

Long-Range Arena (LRA) is a unified benchmark [31] for probing the capability of efficient transformers on handling long sequences. We evaluate our models on five tasks from LRA: ListOps, Text Classification, Retrieval, Image Classification and Pathfinder. All of the tasks are sequence-level multi-class classification. Please refer to the original LRA paper for more details.

Table 5: Experimental results on Long-Range Arena benchmark.

| Model | ListOps | Text | Retrieval | Image | Pathfinder | Avg |
|---|---|---|---|---|---|---|
| Chance | 10.00 | 50.00 | 50.00 | 10.00 | 50.00 | 34.00 |
| Transformer | 36.38 | 64.27 | 57.46 | 42.44 | 88.81 | 57.87 |
| Local Attention | 15.95 | 52.98 | 53.39 | 41.46 | 84.64 | 49.68 |
| Sparse TRans. | 35.78 | 63.58 | 59.59 | 44.24 | 83.90 | 57.42 |
| Longformer | 36.03 | 62.85 | 56.89 | 42.22 | 86.68 | 56.93 |
| Linformer | 35.49 | 53.94 | 52.27 | 38.56 | 86.17 | 53.28 |
| Reformer | 36.30 | 56.10 | 53.40 | 38.07 | 79.18 | 52.61 |
| Sinkhorn Trans. | 34.20 | 61.20 | 53.83 | 41.23 | 73.36 | 52.76 |
| Synthesizer | 36.50 | 61.68 | 54.67 | 41.61 | 81.61 | 55.21 |
| BigBird | 37.08 | 64.02 | 59.29 | 40.83 | 86.75 | 57.59 |
| Linear Trans. | 17.15 | 65.90 | 53.09 | 42.34 | 88.13 | 53.32 |
| Performer | 36.00 | 65.40 | 53.82 | 42.77 | 88.76 | 57.35 |
| Combiner-Fixed | 36.65 | 64.99 | 59.81 | 41.67 | 88.59 | **58.34** |
| Combiner-Axial | 36.15 | 64.36 | 56.10 | 41.33 | 88.43 | 57.27 |

As shown in Table 5, Combiner is able to match the performance of vanilla Transformer and achieves even better performance in some tasks. Following the protocol of LRA, all methods use the same architecture and hyperparameters for a controllable comparison. We use the numbers from Tay et al. [31] for all tasks except for Pathfinder. Since we were unable to reproduce the original Pathfinder results using the default setup in LRA Github repository, we rerun all the baselines using Pathfinder-inter configuration to conduct fair comparison. However, as the benchmark is still of small-scale and the LRA official website discourages hyperparameter tuning, Table 5 should be treated as results for the test bench of expressiveness compared to vanilla Transformer.

Table 6: MLM perplexity on C4 dataset.

| Model | Perplexity |
|---|---|
| Transformer-2k [1] | 4.552 |
| BigBird-2k [41] | 4.696 |
| Performer-2k [28] | 10.940 |
| Fixed-2k [14] | 5.279 |
| Combiner-Fixed-2k (Ours) | 5.170 |
| Axial-2k [20] | 5.370 |
| Combiner-Axial-2k (Ours) | 4.809 |
| Routing-2k [22] | 6.703 |
| Combiner-Routing-2k (Ours) | 6.539 |
| BigBird-8k [41] | 4.542 |
| Combiner-Axial-8k (Ours) | 4.190 |
| Combiner-Fixed-8k (Ours) | **4.139** |

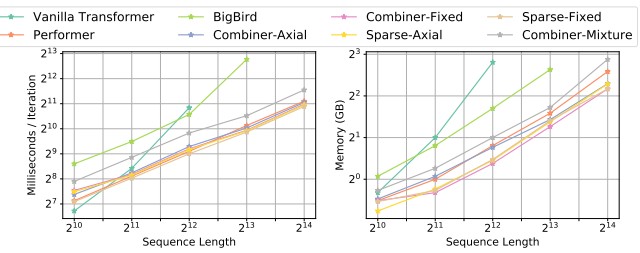

Figure 3: We measure the inference runtime and memory usage for eight models. Overall Combiner has similar speed with Performer and its sparse counterpart but Vanilla Transformer quickly goes OOM when sequence length grows.

### 5.2.2 Masked Language Modeling

As the core element of BERT langauge pretraining [5], masked language modeling (MLM) refers to the task of reconstructing tokens that are randomly masked out in the input sequence. As with the LM task, we use perplexity as the main metric, which correlates relatively well with down-stream task performance. Specifically, we use the large scale C4 dataset [8] for training and evaluation, and consider different sequence lengths. Following the original BERT setup, we mask out 15% of the tokens in each input sequence. The comparison is summarized in Table 6. Similar to the LM result, different Combiner variants consistently outperform their corresponding baselines under 2k sequence length. However, apart from the standard Transformer, Combiner-2k also falls behind BigBird-2k. We conjecture that this is related to the special design in BigBird such as all tokens can always attend to the <cls> token directly, which is only applicable in non-causal problems. That said, when we further increase sequence length to 8k, the standard Transformer runs into OOM issue, whereas Combiner not only outperforms BigBird but also substantially surpasses Transformer-2k. This suggests that Combiner can truly benefit from scaling learning to longer sequence lengths.

### 5.3 Runtime and Memory Usage of Combiner

Here we evaluate the inference runtime and memory usage of five baselines – Transformer, Performer, BigBird, Sparse-Fixed and Sparse-Axial, as well as three variants of Combiner– Combiner-Fixed, Combiner-Axial and Combiner-Mixture. We run inference of all the models on a TPU v3-16 (16 cores x 16GB) with batch size 16, and we test sequences of length from $2^{10}$ to $2^{14}$. As shown in Figure 3, Combiner instantiations achieve comparable runtime and memory usage with their sparse counterpart and Performer. Note Combiner achieves much better empirical performance than the sparse models and Performer. Combiner-Mixture has the same asymptotic complexity with Combiner-Fixed and Combiner-Axial, however, since it requires running two partition plans, it is slower than Combiner-Fixed and Combiner-Axial. Due to the gather operation required by the random attention which is not very TPU/GPU friendly, BigBird is very computationally expensive. And the Transformer model quickly runs out of memory when sequence length increases.

## 6 Conclusion

Inspired by the conditional expectation view of attention mechanism, we propose Combiner, a drop-in replacement of the attention module. By introducing structured decomposition to the conditional probability, Combiner achieves full attention capability while maintaining sub-quadratic computational and memory cost. We instantiate several Combiner variants converting existing sparse transformers to full attention. Combiner achieves state-of-the-art performance on both autoregressive and bidirectional tasks for image and text modeling, showing benefits in both modeling effectiveness and runtime efficiency. Future work includes additional factorization pattern designs, as well as applications of Combiner in domains like bioinformatics and speech.

## Acknowledgments and Disclosure of Funding

We would like to thank Richard Song and David Dohan for the help on introducing Performer codebase and experiment configurations, Yi Tay and Mostafa Dehghani for clarifications on the LRA benchmark, James Lee-Thorp, Joshua Ainslie, and Ilya Eckstein for clarification on their LRA experiment results, Adams Yu for performing internal paper review and helpful suggestions. We also gratefully acknowledge the support of DARPA under Nos. HR00112190039 (TAMI), N660011924033 (MCS); ARO under Nos. W911NF-16-1-0342 (MURI), W911NF-16-1-0171 (DURIP); NSF under Nos. OAC-1835598 (CINES), OAC-1934578 (HDR), CCF-1918940 (Expeditions), IIS-2030477 (RAPID), NIH under No. R56LM013365; Stanford Data Science Initiative, Wu Tsai Neurosciences Institute, Chan Zuckerberg Biohub, Amazon, JPMorgan Chase, Docomo, Hitachi, Intel, JD.com, KDDI, NVIDIA, Dell, Toshiba, Visa, and UnitedHealth Group. Hongyu Ren is supported by the Masason Foundation Fellowship and the Apple PhD Fellowship. Jure Leskovec is a Chan Zuckerberg Biohub investigator.

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
