# Appendix

## A   Universal Approximation

Here we show in Proposition 1 that our Combiner-X achieves universal approximation property [42] if the sparse transformer X achieves universal approximation property. For approaches like BigBird [41], they maintain the universal approximation property using the global tokens (CLS). However, the global attention makes it hard to be applied to the unidirectional autoregressive modeling (LM). Besides, the random attention requires the `gather` operation, making it very slow on dense hardware like TPUs (Figure 3).

**Proposition 1.** *The proposed Combiner will not break the universal approximation property of the original sparse transformers.*

Specifically, we consider the function class constructed by stacking the attention block with a two-layer fully connected network. Formally, following the notations in [42] we have the block as

$$\texttt{SAttn}(X) = X + \texttt{MultiHeadAttn}(X), \tag{13}$$
$$Z = \texttt{SAttn}(X) + \texttt{relu}(\texttt{SAttn} \cdot W_1) \cdot W_2, \tag{14}$$

which denotes the $h$-head attentions with $X \in \mathbb{R}^{L \times d}$, $W_1 \in \mathbb{R}^{d \times r}$, and $W_2 \in \mathbb{R}^{r \times d}$. The function class is denoted as

$$\mathcal{ST}^{H,r} := \{X \to t(X+E) \mid t \text{ is a composition of block (13)}, \tag{15}$$
$$E \text{ is trainable position embedding}\}. \tag{16}$$

Yun et al. [42] shows that the function class (15) is still universal approximation w.r.t. the norm defined as $d_p(f,g) := \left(\int \|f(X) - g(X)\|_p^p \, dX\right)^{1/p}$ with `softmax` in (1) and several requirements on the sparsity patterns in attention scheme.

## B   Combiner-Logsparse in MLM Case

Here we extend the Combiner-logsparse introduced in section 4.2 to the MLM case.

Besides the $\lceil \log_2 i \rceil$ non-overlapping supports in the LM case, we can define addtional $\lceil \log_2 i \rceil$ non-overlapping supports to attend to the tokens after the current token in the sequence. We illustrate this design choice in figure 4.

## C   Combiner-Axial in MLM Case

Besides the $\omega_{\text{axial-vertical}}^{\text{LM}}$, $\omega_{\text{axial-horizontal}}^{\text{LM}}$ and $\omega_{\text{axial-rowmajor}}^{\text{LM}}$ introduced in section 4.3, here we introduce how we extend these three models to the MLM case.

- $\omega_{\text{axial-vertical}}^{\text{MLM}}$: $\Omega_i^0 = \Omega_i^{\text{sparse MLM}} = \{j : j-1 \equiv i-1 (\text{mod } m)\} \cup \{j : j-1 \equiv i-1 (\text{div } m)\}$, and $\Omega_i^r = \{j : j \equiv r (\text{mod } m)\}$, for $r \in [m] \setminus \{col_i\}$. As depicted in Figure 2(A), $\Omega_i^r$ corresponds to the column $r$ above $row_i$, where we use max pooling to obtain the abstraction. To obtain such abstraction for all the locations, we can leverage the `cummax` operator for each column to efficiently obtain the prefix-max.
- $\omega_{\text{axial-horizontal}}^{\text{MLM}}$: similar as $\omega_{\text{axial-vertical}}^{\text{MLM}}$ except that each $\Omega_i^r$ summarizes all rows $r$ and excludes $col_i$.
- $\omega_{\text{axial-rowmajor}}^{\text{MLM}}$: $\Omega_i^0 = \{j : j-1 \equiv i-1 (\text{div } m)\}$, *i.e.*, elements in the same row are directly attended, while $\Omega_i^r = \{j : j \equiv r (\text{div } m)\}$ for $r \in [n] \setminus \{row_i\}$ captures all the rows except $row_i$.

It is trivial to see that the complexity remains $\mathcal{O}(L\sqrt{L})$ if $n, m = \mathcal{O}(\sqrt{L})$.

## D   Combiner-Learnable

As discussed in section 4.4. we design Combiner-learnable as an extension to the routing transformer [22], which learns to cluster the tokens. Each token in the routing transformer only attends to the tokens in the same cluster. As shown in figure 4, our Combiner-learnable combines direct expectation with local expectation (yellow tokens), each of which summarizes one cluster (red, blue or green).

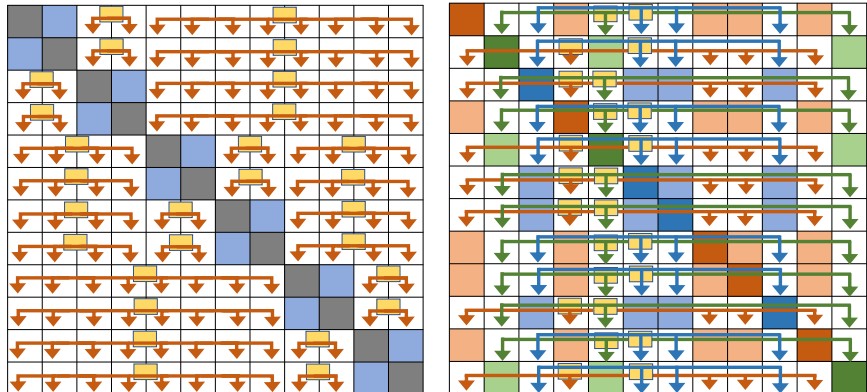

Figure 4: Left: Combiner-logsparse in the MLM case. Right: Combiner-Learnable. Following the routing transformer [22], we apply the combiner principle, so that we can achieve full attention in each head with identical complexity with the routing transformer.

# E Experimental Details

## E.1 CIFAR-10

Here we list the hyperparameters we used on the CIFAR-10 dataset. Our experiments include (1) an ablation study, where all the models share the exact same architecture; and (2) the main result, where our Combiner achieves the state-of-the-art result under the setting that no data augmentation is allowed.

For the ablation study, the embedding and hidden size is 512. We use 8 attention heads in each layer with in total 6 transformer layers. We train all the models for 400,000 steps with learning rate 1e-3 and batch size 32. For the main result, we use the same architecture as introduced in Child et al. [14], and we train our Combiner-Axial for 1,200,000 steps with cosine learning rate scheduling. We rerun the main result for 3 times and the standard deviation is 0.003.

## E.2 ImageNet-64

Regarding the details of the imagenet, we use the same setup with CIFAR-10, which consists of an ablation study and the main result. The architecture used in the ablation study is identical with the one we used in CIFAR-10. For the main result of Combiner-Axial, we used a 30-layer architecture with 768 hidden size and embedding dimension. We train this architecture for 1,200,000 steps with cosine learning rate scheduling. We also rerun the main result for 3 times and the standard deviation is 0.005.

## E.3 Wiki-40B Language Modeling

The main purpose of this experiment is not to chase the state-of-the-art performance, as generally speaking, the more parameters/data, the better the perplexity would be for language modeling. So instead, we let all the methods have the same neural network backbone, while only varying the attention implementations to compare their effectiveness. This is similar in spirit to the ablation study in CIFAR-10 and ImageNet-64.

Specifically, we use the word embedding size and hidden size of 768 for all the layers. We use 12 attention heads in each layer, with in total 12 transformer layers. We use the Pre-Norm architecture, and the MLP layers have hidden size equals to $4 \times 768$. The maximum sequence length can vary in $\{2048, 8192\}$, depends on the memory limit of each methods. All the methods are trained for 125,000 stochastic gradient updates, with batch size equals to 128. We also enable the cosine learning rate scheduling, with 10,000 warm-up steps. The optimizer is Adam with gradient clipping.

## E.4 LRA Benchmark

We mainly follow the guideline of LRA, where all the models should use roughly the same number of parameters and same hyperparameters like batchsize, number of iterations, *etc.*. We tried our best to reproduce the experimental results using the code in https://github.com/google-research/long-range-arena, and we found that we cannot reproduce the `pathfinder-32` results. We have communicated with the authors but didn't get the issue resolved. So instead, we rerun all the baselines using the same network configurations, on the `pathfinder-32-inter` setup. We found some of the methods favor the 'MEAN' pooling to get the sequence representation, while others favor the 'CLS' pooling. So we try both of them for each of the method, and report the best result.

## E.5 C4 Masked Language Modeling

Similar to the purpose of section E.3, we perform masked language modeling task on C4 dataset, which is typically used for BERT pretraining. As the perplexity metric correlates with the downstream task performance well, we thus perform the controlled experiments with all the methods using the same network architecture.

The architecture used and the hyperparameters are almost the same as in section E.3, except that we have maximum number of segments equal 2.