# OpenReview forum: "Combiner: Full Attention Transformer with Sparse Computation Cost"
_NeurIPS.cc/2021/Conference — NeurIPS 2021 Spotlight_

### Official Review · Reviewer_GEmp · 2021-07-01

**Rating:** 7
**Confidence:** 4

**Summary:**

This paper focuses on the problem of reducing the computational cost of attention in transformers. In particular, it addresses the lack of expressiveness in existing attention approaches that leverage sparsity. To this end, the authors propose to view attention as a conditional expectation of embeddings for each token which can be approximated with a structured factorization that includes a direct expectation and an indirect/local expectation term.  Existing sparse attention patterns with sub-quadratic cost can then be used to design such factorizations which result in more expressive attention mechanisms with the same theoretical cost. The evaluation shows that this model can also reach competitive performance in several benchmarks.

**Limitations And Societal Impact:**

The papers discuss theoretical limitations regarding the expressiveness of the proposed factorization which is quite informative. It would be worth including some discussion about the reliance on hand-designed sparsity patterns, the computational cost compared to linear time and space attention methods, and investigate how good speed-quality tradeoffs the proposed models can achieve in practice compared to alternatives. There is no discussion provided about the potential positive or negative societal impact of the work. Although there is no obvious concern, it would be good to see that the authors have thought about it.

**Main Review:**

The paper proposes a new way to factorize attention in transformers using sparse patterns without sacrificing expressiveness.  Even though prior work has achieved linear time complexity attention without sacrificing expressiveness using exponential kernel approximations, the conditional expectation view of attention is a novel contribution that should be of interest in the research community that focuses on efficient transformers.

The proposed framework is theoretically sound and provides a unified view of existing sparse attention patterns and how they can be converted to full attention approximations which is a solid contribution.  The exposition of the structured factorization and its instantiations were clear and comprehensive for the most part. One limitation that was not addressed in the discussion is the computational cost which is bounded by that of the sparse pattern methods, namely sub-quadratic, which is still higher than linear complexity methods. Prior work has shown that both low-rank and full-rank blocks in the approximation are more powerful in theory but it is not clear if it holds up for the proposed approximation since it uses a mixture model to improve expressiveness.  Also, being more powerful also comes at a computational cost; the main point of efficient attention is to achieve a competitive speed-quality tradeoff.  This is an aspect that was not addressed in the model or the evaluation sections (see details below).

The evaluation which is mainly quality-driven shows that the proposed models provide consistent improvements compared to sparse attention methods; the results support the main claim of the paper about improving expressiveness with sparse computational cost. However, the differences between them in terms of speed in practice have not been explored. The proposed models have some additional components that potentially add computational cost in practice. The rest of the evaluation mostly focuses on achieving state-of-the-art results which is great but I think it lacks emphasis on the speed-quality tradeoffs and it lacks an ablation analysis of the components used in the design of the architectures such as Deep Sets for representing the abstraction and the mixture of softmax to obtain a high-rank approximation.  There are also some details that would be helpful for better assessing the results e.g. including the number of parameters for each attention method and reporting the speed achieved by each method in the Long-Range arena benchmark. The only part that involves speed experiments is in Figure 3 which is not thoroughly discussed in the text.

Questions:

- What is the impact of deep sets and the mixture of softmaxes on the performance of the model? An ablation analysis would help to show which component is most important and how good is the approximation without adding any additional parameters.

- The experiment in Figure 3 lacks a detailed description.  One would expect Performer to be faster since it has lower complexity than the proposed models but it's not the case. Could the authors elaborate on that? Does the experiment measure the time it takes to encode or generate sequences of X length? Ideally, we would want to see what happens when generating sequences which is a  challenging setting where parallelization is not possible.

- For the long-range arena, it would be informative to show the runtime for each of the methods so that it is easy to compare methods in terms of speed-quality tradeoffs. Currently, it's not possible to know how good tradeoffs the proposed models achieve in practical tasks.

**Time Spent Reviewing:**

3.5

---

> ### Author Response · Authors · 2021-08-10
> **Response to Reviewer GEmp**
>
> We thank the reviewer for the constructive feedback. Below we clarify important points raised by the reviewer.
>
> **RE ablation study of DeepSets and mixture of softmaxes:**
>
> For the results of DeepSets, we list the experimental results in Table 3 in the general response. We find that DeepSets w/ mean-pooling/max-pooling achieve comparable or better results than Combiner w/o DeepSets. Using max-pooling already yields much better performance than the baselines. One benefit of using max-pooling is that the model has exactly the same number of parameters as the vanilla Transformer / sparse Transformers.
>
> We list the results of ablation study on the mixture of softmaxes in Table 1 in the main paper, where Combiner-Mixture (Ours) represents the method that uses a mixture of softmax with two components: Combiner-Axial-Vertical and Combiner-Axial-Horizontal (Figure 2 in the main paper). Combiner-Mixture has marginally better performance than Combiner-Axial, which only uses one partition/component. We also note that Combiner-Mixture achieves significantly better performance than prior works Reformer, Performer, Routing Transformer with the same architecture (12 layers).
>
>
> **RE linear complexity:**
>
> It is possible to have Combiner variants with linear complexity. We illustrate the possible designs below:
> 1) For bidirectional sequence modeling, we can maintain a constant number of partitions that summarize all locations, and serve as the ‘local expectations’ in Eq (7). As long as the size of $\Omega^0_i$ is constant for all $i$ then we can get linear complexity.
> 2) For unidirectional generative modeling, one can potentially maintain prefixes of local expectations, where each location attends to its prefix summary and current location, while the prefixes can be computed recursively in an overall linear manner similar to RNN.
>
> However, empirically we found that our $O(L \sqrt{L})$ versions are already comparable or even faster than linear ones like Performer, with better quality. Most importantly, the current variants are much easier to implement (see our attached code v.s. BigBird/Performer implementation). Also the current linear variants may have high overhead, or are not TPU friendly (such as the random memory access in BigBird).
>
>
> **RE why faster than Performer:**
>
> Note that although Performer is asymptotically linear, the constant in the computation cost dominates. We follow the default setup in Performer paper to use 256 random features with exp() activation to approximate the softmax kernel. Note that $256^2 = 65,536$, which can potentially make it slower than those $O(L \sqrt{L})$ methods for $L < 65,536$ in our table. Using fewer random features could help, but as it gets inferior quality than Combiner with even 256 random features, we believe our Combiner gets a better speed-quality tradeoff.
>
> **RE time evaluated in Figure 3:**
>
> The figure shows the speed of encoding a sequence. There are several reasons to use encoding for speed comparison:
> 1) Not all the transformer variants (e.g., BigBird) are applicable for generative modeling, thus following LRA protocol we measure the encoding time;
> 2) Though transformers have quadratic computation cost, the main bottleneck is its memory cost (as the computation is parallelizable on TPUs) during encoding. Instead, for generation tasks one can leverage caching to reduce the memory cost to linear, which makes it hard to reveal the fundamental issue with vanilla Transformer.
> 3) It is hard to evaluate the quality of generated text, thus evaluating the perplexity on the held-out data would make it easier to see the time-quality tradeoff.
>
> Note the inference memory and speed are measured on TPU-V3 with batch size 16.
>
> **RE speed-quality tradeoff and comparison between Combiner-X and X:**
>
> We have listed the additional runtime results in Table 2 in the general response. We would like to emphasize that the runtime is not mainly affected by a certain dataset/benchmark, but rather it is mostly affected by the sequence length. Notably we found that our Combiner variants achieved better quality than BigBird, which performs the best in terms of accuracy,  while being empirically faster and more memory efficient than Performer, which achieves the highest efficiency among existing baseline models, in various tasks.
>
> We also show that compared with Sparse-Fixed/Sparse-Axial, our Combiner-Fixed and Combiner-Axial marginally increases the runtime (Table 2 in the general response) while significantly improving the performance (Table 1/3/5 in the main paper). All the results and observations demonstrate that Combiner can achieve a good speed-quality tradeoff.

---

> > ### Comment · Reviewer_GEmp · 2021-08-11
> > **Thanks for the response**
> >
> > The additional results and clarifications answered most of my questions; it'd be great if they are reflected in the final version. A few comments regarding the replies:
> > - It was not clear in the text that Table 1 addresses this particular question. The point about the performance brought by MoS is not discussed at all in the results. Also, if MoS represents the best model in Table 1 what is the added value from Combiner?
> > - The point about caching does not seem correct. Leveraging caching during decoding reduces the complexity to a linear **per step** which means that the cost is still quadratic with respect to the sequence length (e.g. see https://openreview.net/pdf?id=KpfasTaLUpq).
> > - To evaluate the speed-quality tradeoff of the models on generation tasks one can use machine translation as a task. The fact that some of the existing models are restricted to encoding-only tasks does not make generation tasks less important. It'd be useful to explain this rationale in the paper and clarify the scope of the findings because the efficiency improvements are not necessarily applicable during generation (currently, Figure 3 is not even cited in the main text).

---

> > > ### Author Response · Authors · 2021-08-17
> > > **Further clarifications to reviewer GEmp**
> > >
> > > Thank you for the additional questions. We’ve done additional experiments to make clarifications. Specifically:
> > >
> > > -  **RE: concerns about MoS**
> > >
> > > The “MoS” or Mixture of softmaxes corresponds to a variant in the Combiner family that uses a mixture model to approximate the conditional distribution (also referred as Combiner-Mixture), and is different from the original MoS paper which increases the expressiveness in the output layer. We would like to emphasize that the Combiner-Mixture also belongs to our contribution. It comes with the tradeoff of expressiveness/speed+memory (see Table 1 and 2 in the general response).
> > >
> > > Or if the reviewer is concerned about the ablation of the mixture idea itself, one can definitely combine different sparse patterns to form a more powerful approximation. BigBird is such an example that mixes local+global+random patterns. However it achieves worse trade-off than Combiner (see Table 1,2 above, as well as the numerical results in the main paper).
> > >
> > > Let us know if the above reasoning resolves your concern.
> > >
> > > - **RE: caching for vanilla Transformer**
> > >
> > > Our above response is about the **memory cost** of the vanilla Transformer. Though the computation cost is still quadratic during sequence generation, the memory cost is **linear with caching**. To achieve this, one can make a buffer to keep the historical Q,K,V matrices in each layer. Each step only involves attention from the current location w.r.t the cached Q,K,V, and the size of these matrices will grow by 1 row after the generation of the new token. To make an analogy, the LSTM has $O(L)$ complexity, but it only requires $O(1)$ memory during sequence generation.
> > >
> > > We’ve provided a PyTorch implementation to help better understand this. The link to the anonymous code can be found below.
> > >
> > > **https://drive.google.com/file/d/1vO7L61AiwsojVGKW13wsCIR8J4skE_MD/view**
> > >
> > > We’ve also experimentally verified below that the vanilla Transformer can indeed generate super long sequences (e.g., $L = 2^{19}$).
> > >
> > > - **RE: evaluation on the generation tasks**
> > >
> > > We carried out additional experiments as suggested by the reviewer to compare the sequence generation speed between vanilla Transformer and Combiner-Fixed.
> > >
> > > Below are the runtime comparisons on different devices.
> > >
> > > [**Figure: generation speed on CPU**](https://i.ibb.co/sq5mQN3/decode-cpu.png)
> > >
> > > [**Figure: generation speed on GPU**](https://i.ibb.co/xjbrC3D/decode-gpu.png)
> > >
> > > To summarize our findings:
> > >
> > > 1. Combiner runs faster on generation tasks when the sequence gets long enough.
> > > 2. Unlike in training, the speed-up during generation won’t show up until the sequence gets long enough (e.g., longer than $2^{16}$ on GPU). This is because the generation cost per step is $O(L)$ for Transformer and $O(\sqrt{L})$ for Combiner. As the per-step operation is parallelizable (e.g., attending to all previous locations in vanilla transformer requires $O(L)$ independent vector products and $O(1)$ reduction operations), despite that it is serial across steps, $O(L)$ and $O(\sqrt{L})$ cost won’t make a difference until $L$ exceeds the parallel power of hardware (e.g., larger than the number of CUDA cores).
> > > 3. Transformer can generate much longer sequences than what it can be trained on. So the bottleneck is in training.
> > >
> > > We will incorporate all the above text and new results into the final version of our paper.

---

> > > > ### Comment · Reviewer_GEmp · 2021-08-18
> > > > **Response**
> > > >
> > > > Thanks for the great effort and the clarifications! The experiments and discussion about efficiency aspects will definitely add value because they were somewhat lacking. I am still positive about the paper but my concerns have not been fully resolved:
> > > >
> > > > - Using MoS to approximate the conditional distribution of attention is part of the contribution, I agree. However, MoS in my view is an orthogonal approach to the proposed approximation that could also be applied to competitor approximations. The increase in representational capacity brought by MoS and how varying its components affect Combiner-Mixture's performance is not very clear. For instance, what is the performance of Combiner-Mixture when a single component is used? It would be helpful for the reader to have more transparency about the selection process for the number of components and their impact on performance.
> > > >
> > > > - Got it, thanks. My intent was to emphasize the computational cost aspect because I was not convinced about the arguments in favor of evaluating only in terms of encoding speed.  The comment "one can leverage caching to reduce the memory cost to linear, which makes it hard to reveal the fundamental issue with vanilla Transformer" came across as if the quadratic cost in transformers doesn't matter because memory cost is linear. During encoding, it's true that memory is the only bottleneck but this is not the case during decoding. Performing caching will not solve the computational problem for long sequences that causes decoding latency, so it is not clear what the authors are trying to prove with this toy experiment. It is not surprising being able to generate very long sequences with vanilla transformers if one can fit it into memory with caching; the point is how slow the model is going to be.  Lastly, focusing on the speed metric alone without any indication of the quality for the generation task is not very informative but I appreciate the efforts.

---

> > > > > ### Author Response · Authors · 2021-08-21
> > > > > **Response to reviewer GEmp**
> > > > >
> > > > > Thank you for the additional questions. We address your concerns below:
> > > > >
> > > > > > For instance, what is the performance of Combiner-Mixture when a single component is used?
> > > > >
> > > > > We have **already conducted this ablation study in the very first submission**, please check the Table 1 in the main paper, where *Combiner-Axial* is equivalent to *Combiner-Mixture with a single component*. We will add clarifications to the paper.
> > > > >
> > > > > > Performing caching will not solve the computational problem for long sequences that causes decoding latency, so it is not clear what the authors are trying to prove with this toy experiment
> > > > >
> > > > > We want to prove that Combiner enjoys significant speedup in **both encoding and decoding**. For encoding, please check the Figures in the [**general response**](https://openreview.net/forum?id=MQQeeDiO5vv&noteId=-h5HnwArwV-). For decoding, please check the Figures in [**Further Clarifications to reviewer GEmp**](https://openreview.net/forum?id=MQQeeDiO5vv&noteId=kqfbFawEk5T).
> > > > >
> > > > > > My intent was to emphasize the computational cost aspect because I was not convinced about the arguments in favor of evaluating only in terms of encoding speed.
> > > > >
> > > > > Thank you for the concern. This is exactly what we try to address in [**Further Clarifications to reviewer GEmp**](https://openreview.net/forum?id=MQQeeDiO5vv&noteId=kqfbFawEk5T), where we have demonstrated that Combiner would also have faster generation speed than the vanilla Transformer.

---

> > > > > > ### Comment · Reviewer_GEmp · 2021-08-23
> > > > > > **Response to authors**
> > > > > >
> > > > > > The reply does not actually address my concerns. Here is why:
> > > > > >
> > > > > > The question about what is the effect of the number of MoS components to the performance of the model is neither addressed in your replies  nor in the paper. Two data points in Table 1 without explanations or discussion does not constitute an "ablation study" about this issue in my view. A simple clarification of how many components correspond to these two models doesn't address the question above.
> > > > > >
> > > > > > The evaluation would require much more effort to be convincing about any decoding speed claim. Measuring decoding speed without an actual task and performance metric in the experimental design is not how prior work has evaluated efficient transformers in the area and does not reflect expected speed-quality tradeoffs in real world tasks.

---

> > > > > > > ### Author Response · Authors · 2021-08-24
> > > > > > > **Further Response to reviewer GEmp**
> > > > > > >
> > > > > > > Thank you for the follow up to further clarify your concerns. See our response below:
> > > > > > >
> > > > > > > > what is the effect of the number of MoS components…
> > > > > > >
> > > > > > > We show that a single component is good enough in most cases (see all the tables except Table 1 in the main paper). Table 1 in the main paper shows that 2 components can improve the quality further but also with the increase in memory. Given that TPU-v2 only has 8G per core, we believe the signal of quality/memory trade-off here is clear.
> > > > > > >
> > > > > > >
> > > > > > > > … is not how prior work has evaluated efficient transformers in the area …
> > > > > > >
> > > > > > > We found that most of the efficient transformers only evaluate the training speed (or speed for encoding), specifically:
> > > > > > > - [**Sparse Transformer**](https://arxiv.org/pdf/1904.10509.pdf) measured the speed per iteration in their Table 2.
> > > > > > > - [**Reformer**](https://arxiv.org/pdf/2001.04451.pdf) measured seconds per training step in their Figure 5.
> > > > > > > - [**Routing Transformer**](https://arxiv.org/pdf/2001.04451.pdf) measured the steps per second in their Table 7.
> > > > > > > - [**Performer**](https://arxiv.org/pdf/2009.14794.pdf) measured the forward and backward time separately in their Figure 3.
> > > > > > > - [**LRA benchmark**](https://openreview.net/forum?id=qVyeW-grC2k) measured the encoding time in Table 2.
> > > > > > > - [**Linformer**](https://arxiv.org/pdf/2006.04768.pdf) evaluated the forward time during inference in their Table 3.
> > > > > > >
> > > > > > > Meanwhile, some works didn’t provide explicit empirical speedup, but rather show the scalability via asymptotic analysis or the metric from long sequence benchmarks, including:
> > > > > > > [**Logsparse Transformer**](https://arxiv.org/pdf/1907.00235.pdf), [**Bigbird**](https://arxiv.org/pdf/2007.14062.pdf),  [**Axial Transformer**](https://arxiv.org/pdf/1912.12180.pdf), etc.
> > > > > > >
> > > > > > > We believe it is fair to follow the most popular protocol to show the scalability, where in the paper we have provided *both* the *asymptotic* and *empirical encoding* speed up.
> > > > > > >
> > > > > > > Though generation speed is not our main focus, we show that Combiner can generate faster than Transformer, as requested by the reviewer. We follow the [**Linear Transformer**](https://arxiv.org/pdf/2006.16236.pdf) Figure 1 to use synthetic tasks to better see the scale up w.r.t different sequence length, as the runtime mainly depends on the length and architecture.
> > > > > > >
> > > > > > > Again, we show the possibility of faster generation mainly in response to the reviewer's concern. How to speed up the generation of autoregressive models is another bigger topic and is not the focus of our contribution. We believe that the evaluation of runtime and memory in Table 1 and 2 in the general response has already followed “*how prior work has evaluated efficient transformers in the area*” and has already demonstrated the scalability of Combiner.

---

> > > > > > > > ### Comment · Reviewer_GEmp · 2021-09-01
> > > > > > > > **Response to authors**
> > > > > > > >
> > > > > > > > Thanks for clarifying the scope, it sounds reasonable. I might have overestimated the importance of MoS here but I still think it's worth discussing its effect and providing insights about it.

---

### Official Review · Reviewer_4mpH · 2021-07-09

**Rating:** 7
**Confidence:** 3

**Summary:**

Combiner - a new cheaper computationally attention algorithm. By introducing a new factorization structure, the design an algorithm with low complexity but capability close to full attention.
Numerous experiments demonstrate the utility of the proposed approach.

**Ethical Concerns:**



**Limitations And Societal Impact:**

see above comments for approximation properties of the Combiner

**Main Review:**

The topic of approximate attention is a valid question.
The current paper seems to have a very performant approach to that, and their experimental results seem solid.
However, I am concerned that some of the mechanics of the method are not fully explained, and too strong claims are made. Some revisions could make it better.

- l. 5 "full attention capability" l.132 "full attention model almost for free" - I am puzzled by the claims that the method is equivalent to full attention, while other sections of the paper say it is an approximation. Please clarify this exactly.

- l.123 subquadratic cost. I do not understand fully the claim. In general, if A(x_i) is attention for one query position i, and there are L such positions i. To calculate each A(x_i) in equation (8) one sums over all j positions, which are also L --> it seems to me still quadratic.

- also, please discuss how the choice of n_i, the partition size and count, affects the computation costs and model capacity of the Combiner

- section 3.3 tradeoffs - please improve and  clarify that section, my above comments are also concerned with better understanding of the factorization and its expressiveness

- l.150 I do not understand how exactly max pooling is applied, is it per dimension d of the keys k_j, or the most active key overall, or some other way to do it?

- reference [40] BigBird seems very related to the Combiner. While it is good there is an empirical comparison with it, would be good to mention it also in the introduction related works.  The claims of both papers (efficient attention that keep the properties of full attention due to complicated patterns) seem related to me, so would be good to elaborate on that. I saw that the appendix has some mentions on both papers and the "universal approximation property", but it did not answer fully my questions for the comparison of the two methods.

- Experiments, in table 4 the Combiner is even better than full attention, please discuss this, e.g. mention whether this is due to some inductive bias or other reason?


**Time Spent Reviewing:**

5

---

> ### Author Response · Authors · 2021-08-10
> **Response to Reviewer 4mpH**
>
> We thank the reviewer for the constructive feedback. Below we clarify important points raised by the reviewer.
>
> **RE clarification of several terminology in the paper:**
>
> By *full attention capacity*, we mean each token is able to attend to all the previous tokens (unidirectional) or all the tokens in the sequence (bidirectional), in the same probabilistic sense (i.e., conditional expectation of value vectors) as vanilla Transformer. Our combiner can achieve this property, in contrast to the existing sparse transformers, in which the model only attends to a portion of the tokens. By *full attention almost for free*, it means that the asymptotic complexity of Combiner is exactly the same as various sparse transformers. By *approximation*, we mean our Combiner exploits a factorized conditional expectation in Eq. (6) to approximate the original conditional distribution in Eq. (3). This leads to a locally low-rank but globally high-rank approximate attention matrix, please find more details and discussions in Section 3.3.
>
> **RE sub-quadratic cost:**
>
> Indeed for each position $i$, $A(x_i)$ covers all $L$ positions in the sequence, however, our combiner achieves sub-quadratic cost with the key insight being the position $i$ attends to $L$ positions using several "proxies". Concretely, for a location $i$, assume we have $n_i + 1$ proxies, $\Omega_i^0, \dots, \Omega_i^{n_i}$, and each $\Omega_i^j$ covers a subset of tokens, with $\Omega_i^j \neq \Omega_i^k, j\neq k$ and $\cup_j \Omega_i^j = [L]$. The idea is that each token $i$ will directly attend to all locations within $\Omega_i^0$ (the direct expectation term in Eq. (7)) as well as the proxies $\Omega_i^1, \dots, \Omega_i^{n_i}$ (the $p(\Omega_i^r|i)$ term in Eq. (7)). At the same time each proxy $\Omega_i^r, r\in[1, n_i]$ will "summarize" all the positions that belong to the proxy (the local expectation term in Eq. (7)).
>
> If we analyze the cost of the three terms,
> 1) direct expectation equals the number locations in $\Omega_i^0$: $\sum_{i=1}^L |\Omega_i^0|$;
> 2) $p(\Omega_i^r|i), \forall r\in [1, n_i]$ equals $n_i$: $\sum_{i=1}^r n_i$;
> 3) local expectation equals the total size of all distinct proxies combined: $|\{\Omega_i^r\}_{:i\in[L], r\in[1,n_i]}|$.
>
> For example, in Combiner-fixed, we have
> 1) $|\Omega_i^0|$ is $\mathcal{O}(\sqrt{L})$, then overall being $\mathcal{O}(L \sqrt{L})$;
> 2) $n_i$ is $\mathcal{O}(\sqrt{L})$, then overall being $\mathcal{O}(L \sqrt{L})$;
> 3) together across all positions we have $\sqrt{L}$ different proxies, the size of each proxy equals $\sqrt{L}$ (the size of a row), then the overall complexity for term (3) is $O(L)$.
>
> Together the complexity is $\mathcal{O}(L \sqrt{L})$, and we achieve the full attention capacity.
>
> **RE please discuss how the choice of n_i, the partition size and count, affects the computation costs and model capacity of the Combiner:** please find the discussion in the above paragraph and section 3.1 and all section 4 in much detail.
>
> **RE section 3.3 tradeoffs:** We will add the above examples and texts to the paper to better clarify the tradeoffs.
>
> **RE max-pooling:** The reviewer raises a question on how the max-pooling is performed in Combiner. $k_{\Omega_i^r}={MaxPooling}_{j}$   $k_j$, max pooling is applied across each dimension.
>
> So the dimension of $k_{\Omega_i^r}$ is the same as each $k_j$.
>
> **RE comparison with BigBird:** Our combiner has two key improvements and benefits compared with BigBird:
> 1) BigBird requires the global CLS tokens for full attention, however, it cannot be applied to unidirectional causal sequence modeling setting, while our combiner can be widely applied to many sparse transformers in both causal (unidirectional) and masked (bidirectional) sequence modeling;
> 2) BigBird requires the gather operation in the Random Attention component, which is expensive for dense hardwares such as TPUs. However, our Combiner-Axial / Combiner-Fixed is very TPU/GPU friendly as shown on Figure 3 in the main paper for speed and Table 1 in the author response for memory consumption. We will add the above discussion about BigBird in our final version.
>
> **RE performance on LRA:** In fact, there are also many methods that achieve better performance than the vanilla Transformer on various tasks in LRA, e.g., BigBird and Synthesizer on ListOps, Linear Transformer and Performer on Text, Sparse Transformer and BigBird on Retrieval, Performer on Image. We argue the tasks in the benchmark are still small-scale (since vanilla Transformer still runs) and the goal is to show Combiner achieves comparable performance with vanilla Transformer in downstream tasks. Since the LRA github repository explicitly discourages hyperparameter tuning, this benchmark should be treated as ‘testbench’ rather than a competition.
>
> We will incorporate the above texts and add more detailed description and clarification in the final version.

---

> > ### Comment · Reviewer_4mpH · 2021-08-11
> > **response to author rebuttal**
> >
> > thanks a lot for your detailed answer, it clarified it for me.

---

### Official Review · Reviewer_ETpX · 2021-07-16

**Rating:** 9
**Confidence:** 4

**Summary:**

This paper proposes the combiner, which could achieve full attention capability while maintaining sub-quadratic computational and memory cost.

**Limitations And Societal Impact:**

Yes, this method keeps the full attention capability while maintaining low computation and memory complexity.

**Main Review:**

Originality: This paper is pretty novel.

Clarity: This article is very clear and the formula derivation is detailed and rigorous.

Significance: For a long time, self-attention has been computationally cumbersome, especially when dealing with some very long sequences. This article makes self-attention more efficient while ensuring expressive ability. The theoretical derivation in this article is very sufficient, and the experiments are also very solid.

Question: I am curious about the comparison between this paper with the spatial linear layer[1,2]. Could this method be more efficient?

[1]. Synthesizer: Rethinking Self-Attention for Transformer Models
[2]. Pay attention to MLPs

**Time Spent Reviewing:**

2 hours

---

> ### Author Response · Authors · 2021-08-10
> **Response to Reviewer ETpX**
>
> We thank the reviewer for the constructive feedback. Below we clarify important points raised by the reviewer.
>
> **RE relationship between Combiner and spatial linear layer:**
>
> Both Synthesizer and gMLP are trying to replace the “adaptive attention matrix” with a trainable parameter matrix, which still has a **quadratic dependency** on the sequence length. Hence, they are not trying to solve the efficiency problem that we are concerned with here. Instead, they are exploring other types of spatial interaction beyond attention. We will provide detailed discussion in our final version.

---

> > ### Comment · Reviewer_ETpX · 2021-09-10
> > **Reply to Authors' Responses**
> >
> > Thanks for your clarification.

---

### Official Review · Reviewer_u2iY · 2021-07-19

**Rating:** 8
**Confidence:** 4

**Summary:**

This paper presents combiner, a new variant of sparse Transformer, which is inspired by viewing self-attention as a conditional expectation over embeddings at each location. The authors show that combiners achieve better performance than existing sparse Transformer variants.

**Limitations And Societal Impact:**

The main limitation of the paper is that the real overhead using the proposed methods is not fully revealed (see the weaknesses part above).

**Main Review:**

Strengths:
1. The proposed method is novel and simple. It is surprising to see that simple max-pooling already produces strong performance.
2. The paper is well written and easy to follow.
3. It is great to see some significant improvement over the previous methods on a wide range of tasks.

Weaknesses:
1. It would be great if we could see a table containing the memory footprint and inference speed results following the setup in Table 2 in the Long Range Arena paper. Otherwise, it is unclear whether the proposed method saves the memory as well. Also, having such a table provides the numbers for future works to compare with.
2. The authors only time the speed of Combiner-Axial and Combiner-Fixed. The speed overhead of Combiner-Mixture is unrevealed.
3. It is unclear to me how many partitions and what they are in the Combiner-Mixture model in Table 1.
4. The proposed Combiner-Logsparse is not evaluated in the experiment section. I wonder what the reason is.
5. On some tasks Combiner-Axial outperforms Combiner-Fixed, but on some others, it doesn't. This makes it hard for the practitioners to decide which to use.
6. It would be better to see an ablation study on different alternatives to max-pooling when combining queries and keys. For example, how do DeepSets mentioned in subsection 3.2 perform?


Overall, the proposed method is original, interesting, and effective, so I recommend accepting it. Admittedly, if the authors can address the weaknesses above (at least the first four) in their rebuttal, I will raise my rating to 8.

**Time Spent Reviewing:**

3.5

---

> ### Author Response · Authors · 2021-08-10
> **Response to Reviewer u2iY**
>
> We thank the reviewer for the constructive feedback. Below we clarify a number of important points raised by the reviewer.
>
> **RE memory:**
>
> We have attached the table of the memory usage of different methods in Table 1 in the general response. Since LRA did not release the code to reproduce the runtime/memory numbers, and it matters a lot if using different frameworks/platforms, we compared the model with the best performance in LRA (BigBird) and the model with the highest efficiency (Performer, according to Table 2 in LRA) in our codebase. It shows that given the same sequence length, Combiner has much smaller memory consumption than vanilla Transformer and BigBird, and is even more memory efficient than Performer because Performer uses 256 random features (which is the default setting in the Performer paper) and 256 is larger than square root of the maximum length 16k (the $\sqrt{L}$ term in Combiner-Fixed and Combiner-Axial). Inference memory and speed are measured on TPU-V3 with batch size 16. Details of these tests will be included in the appendix.
>
> **RE clarification of Combiner-Mixture:**
>
> The Combiner-Mixture model has two partition plans/components in the mixture model. Here the two plans correspond to the Combiner-Axial-Vertical and the Combiner-Axial-Horizontal (as shown in Figure 2 in the main paper). When calculating the final $A(x_i)$, we average the conditional distribution calculated by the two components (details in line 174-180). We also attached the runtime of Combiner-Mixture in Table 2 in the general response. Combiner-Mixture has much faster runtime than BigBird and vanilla Transformer, and did not run out of memory for 16k sequence length. Given a tight memory/runtime budget, we can safely use Combiner-Axial and Combiner-Fixed for comparable performance but less memory/runtime than Combiner-Mixture. We will add the tables to the paper.
>
> **RE evaluation of Combiner-Logsparse and discussion on Combiner-Axial / Combiner-Fixed:**
>
> The reviewer raises a question on the evaluation of Combiner-Logsparse and how to select different Combiner variants. First, we evaluated Combiner-Logsparse in the ablation study on CIFAR-10 and ImageNet-64. As shown in Table 1 in the main paper, given the same architecture, Combiner-Logsparse performs much better than its sparse counterpart Logsparse transformer [18]. We reemphasize that our contribution is to propose a general framework to grant prior sparse transformer models with full attention capability while keeping the exact same asymptotic complexity. We show that all three sparse transformers can be greatly benefited from using the Combiner idea. Choosing the model really depends on the data and task. Moreover, Combiner is flexible enough so that it can be further applied to future works if a new and more effective sparse pattern is proposed.
>
> **RE DeepSets:**
>
> The reviewer wonders how the DeepSets variant performed. As shown in Table 3 in the general response, we report the results of several variants of Combiner-Fixed that use DeepSets w/ mean-pooling or max-pooling respectively to summarize the $q_{\Omega_i^r}$ and $k_{\Omega_i^r}$ (section 3.2). Using DeepSets achieves marginally better performance. And they all yield significant improvement from baselines. We note the benefit of directly using max-pooling rather than DeepSets is that it **has the exact same number of parameters** as the vanilla transformer models.

---

> > ### Comment · Reviewer_u2iY · 2021-08-10
> > **Reply to Authors' Responses**
> >
> > The authors' responses have addressed my concerns on the memory usage and inference time on LRA as well as the alternative of using Deepset. Therefore, I have updated my rating from 7 to 8.

---

### Author Response · Authors · 2021-08-10
**General Response with New Results**

We thank all the reviewers for their invaluable feedback. Overall, we are glad that the reviewers found Combiner to be a novel contribution to advance efficient transformers. Below we add clarifications and new results as requested by the reviewers. Reviewers suggest we report the memory footprint (Table 1) and the speed (Table 2) of baselines and several variants of our Combiner. These tables are also plotted into Figures

- [**Figure: memory**](https://i.ibb.co/wrRCCvz/mlm-memory-rebuttal.png)

- [**Figure: speed**](https://i.ibb.co/JddfJ09/mlm-speed-rebuttal.png)

following Figure 3 in the main paper. Overall Combiner-X achieves much better performance than the sparse baseline X (Table 1/3/5 in the main paper) with comparable memory and runtime. Besides, we also report the performance of Combiner-DeepSets on LM-Wiki40b dataset. We show that Combiner-DeepSets performs marginally better than Combiner.

| Memory (GB)  |  1k  |  2k  |  4k  |  8k  |  16k |
|------------------|:----:|:----:|:----:|:----:|:----:|
| Vanilla          |  0.8 |   2  |   7  |  OOM |  OOM |
| Performer        |  0.7 |   1  | 1.75 |   3  |   6  |
| BigBird          | 1.05 | 1.75 | 3.25 |  6.2 |  OOM |
| Sparse-Fixed     | 0.69 | 0.83 | 1.39 | 2.65 |  4.5 |
| Combiner-Fixed   |  0.7 |  0.8 |  1.3 |  2.4 |  4.5 |
| Sparse-Axial     | 0.59 | 0.85 | 1.37 |  2.6 | 4.88 |
| Combiner-Axial   | 0.72 | 1.05 |  1.7 |  2.7 |  4.9 |
| Combiner-Mixture | 0.83 |  1.2 |   2  |  3.3 | 7.35 |
Table 1. Peak memory usage of Vanilla Transformer, Performer, BigBird, and Sparse Transformers and their Combiner counterpart on masked language modeling with different sequence lengths.

| Inference Speed (ms in log base 2) |  1k  |  2k  |   4k  |   8k  |  16k  |
|-----------------------------------|:----:|:----:|:-----:|:-----:|:-----:|
| Vanilla                           | 6.72 | 8.42 | 10.84 |  OOM  |  OOM  |
| Performer                         | 7.13 | 8.11 |  9.10 | 10.13 |  11.1 |
| BigBird                           |  8.6 | 9.49 | 10.57 | 12.77 |  OOM  |
| Sparse-Fixed                      | 7.49 | 8.15 |  9.17 |  9.92 | 10.95 |
| Combiner-Fixed                    | 7.54 | 8.18 |  9.2  |  9.94 | 10.98 |
| Sparse-Axial                      | 7.09 | 8.03 |  9.0  |  9.87 | 10.89 |
| Combiner-Axial                    | 7.38 | 8.24 |  9.29 | 10.03 | 11.05 |
| Combiner-Mixture                  |  7.9 | 8.86 |  9.83 | 10.52 | 11.55 |
Table 2. Inference runtime (ms in log base 2) of Sparse Transformer and their Combiner counterpart and Combiner-Mixture following the setup of Figure 3.

| LM-Wiki40b (sequence length)                      | Perplexity |
|----------------------------------|:----------:|
| Combiner-Fixed-DeepSets-Max (8k)  |    16.29   |
| Combiner-Fixed-DeepSets-Mean (8k) |    16.48   |
| Combiner-Fixed-Max (8k)          |    16.6    |
| Combiner-Fixed-Mean (8k)         |    16.54   |
| Vanilla Transformer (2k)         |    17.26   |
Table 3. DeepSets variant of Cominer with Mean/Max pooling on LM-Wiki40b. We show that Combiner-DeepSets achieves marginally better performance than Combiner without DeepSets (8k window size). Both achieve much better performance than vanilla transformer (2k window size)

---

### Decision · Program_Chairs · 2021-09-27

**Decision:**

Accept (Spotlight)

**Comment:**

This paper proposes a novel efficient Transformer variant, called Combiner, which is a drop-in replacement of attention, achieving full attention with sub-quadratic cost using structured factorization. The reviewers have several concerns regarding the experimental analysis in the initial review. During the rebuttal period, the authors provided intensive analysis and addressed most of the concerns from the reviewers.

All reviewers recommend acceptance, and I hope the authors could include the new experimental results and ablation studies in the camera-ready version.